# Effects of Juvenile or Adolescent Working Memory Experience and Inter-Alpha Inhibitor Protein Treatment after Neonatal Hypoxia-Ischemia

**DOI:** 10.3390/brainsci10120999

**Published:** 2020-12-17

**Authors:** Aaron Bradford, Miranda Hernandez, Elaine Kearney, Luke Theriault, Yow-Pin Lim, Barbara S. Stonestreet, Steven W. Threlkeld

**Affiliations:** 1Neuroscience Program, School of Health Sciences, Regis College, 235 Wellesley Street, Weston, MA 02493, USA; aaron.bradford@regiscollege.edu (A.B.); mher760@regiscollege.edu (M.H.); ekea983@regiscollege.edu (E.K.); lthe540@regiscollege.edu (L.T.); 2ProThera Biologics, Inc., 349 Eddy Street, Providence, RI 02903, USA; Yplim@protherabiologics.com; 3Department of Pathology and Laboratory Medicine, The Alpert Medical School of Brown University, 222 Richmond Street, Providence, RI 02903, USA; 4Department of Pediatrics, The Alpert Medical School of Brown University, Women & Infants Hospital of Rhode Island, 101 Dudley Street, Providence, RI 02905, USA; BStoneStreet@wihri.org

**Keywords:** hypoxia-ischemia, encephalopathy, Inter-alpha Inhibitor Proteins, immune modulator, inflammation, eight-arm water maze, early behavioral intervention, working memory

## Abstract

Hypoxic-Ischemic (HI) brain injury in the neonate contributes to life-long cognitive impairment. Early diagnosis and therapeutic interventions are critical but limited. We previously reported in a rat model of HI two interventional approaches that improve cognitive and sensory function: administration of Inter-alpha Inhibitor Proteins (IAIPs) and early experience in an eight-arm radial water maze (RWM) task. Here, we expanded these studies to examine the combined effects of IAIPs and multiple weeks of RWM assessment beginning with juvenile or adolescent rats to evaluate optimal age windows for behavioral interventions. Subjects were divided into treatment groups; HI with vehicle, sham surgery with vehicle, and HI with IAIPs, and received either juvenile (P31 initiation) or adolescent (P52 initiation) RWM testing, followed by adult retesting. Error rates on the RWM decreased across weeks for all conditions. Whereas, HI injury impaired global performance as compared to shams. IAIP-treated HI subjects tested as juveniles made fewer errors as compared to their untreated HI counterparts. The juvenile group made significantly fewer errors on moderate demand trials and showed improved retention as compared to the adolescent group during the first week of adult retesting. Together, results support and extend our previous findings that combining behavioral and anti-inflammatory interventions in the presence of HI improves subsequent learning performance. Results further indicate sensitive periods for behavioral interventions to improve cognitive outcomes. Specifically, early life cognitive experience can improve long-term learning performance even in the presence of HI injury. Results from this study provide insight into typical brain development and the impact of developmentally targeted therapeutics and task-specific experience on subsequent cognitive processing.

## 1. Introduction

Perinatal Hypoxic-Ischemic (HI) brain injury is associated with lack of oxygen and blood flow (hypoxia and ischemia, respectively) to the brain, often resulting in rupture to blood vessels within the highly vascularized subependymal germinal matrix in infants [1]. HI-related brain injury is characterized by patterns of subsequent necrosis, cavitation, and continuing gliosis, although surrounding regions may survive and undergo neuronal plasticity [2]. Hypoxic-ischemic encephalopathy (HIE) is a defined clinical disorder that occurs in 3–5 per 1000 full term infants [3,4]. However, premature infants can also be exposed to HI related brain injury [3,4]. White matter and cortical tracts may show lasting damage depending on the timing and severity of the injury [3,5]. Currently, hypothermia is the only available therapeutic strategy to treat full term infants with HIE [6]. This therapy has not been completed in trials extending clinical treatment to premature infants exposed to HI-related brain injury. Consequently, the only treatment available for premature infants is supportive care.

Therapeutic hypothermia can improve survival, reducing severe physical and intellectual impairment including cognitive delays, cerebral palsy, or death in an estimated 40–55% of infants with HIE [7,8,9,10,11]. However, HI-related brain injury requiring therapeutic hypothermia might represent the most severe form of perinatal brain injury, whereas milder forms of HI and those without sentinel events may not be detected [6]. Therefore, improvement in diagnostics, development of alternative treatment strategies, and improved assessment of the life-long impact of HI related brain injuries are essential.

The ability of hypothermia to improve survival after HIE strongly suggests that the severity of injury could be reduced by other anti-inflammatory strategies. Specifically, hypothermia reduces the overall systemic inflammatory and metabolic responses to injury in addition to having specific neuroprotective effects [12,13,14,15,16]. Therefore, other anti-inflammatory approaches may offer clinically valid alternatives or adjunctive therapeutic effects to hypothermia for the treatment of HIE [17,18,19]. Inter-alpha Inhibitor Proteins (IAIPs) are a family of plasma proteins found abundantly in mammals including rats, sheep, and humans and are thought to play a significant role in regulating inflammation [20,21,22,23,24,25]. Administration of IAIPs and their metabolites has been investigated in multiple inflammatory disorders, including sepsis and the destructive inflammatory responses to SARS-CoV-2 [25], and has shown systematic protective effects in rodents via reduction in complement activation and other pro-inflammatory mechanisms [26,27,28]. Experimental HI-related brain injury can be induced in rodents via carotid ligation and exposure to a hypoxic environment [29]. This model for perinatal HI injury has been used to assess multiple pathophysiological and behavioral factors that can be compared to human outcomes including inflammatory responses [30], structural anomalies [31], and cognitive processes including deficits in working memory [32]. Administration of IAIPs facilitate significant neuroprotective effects after exposure to HI-related brain injury [33,34,35]. These studies confirm the systematic anti-inflammatory effects of IAIP administration at the sites of HI-related brain injury, which appear to down regulate reactive gliosis markers, infiltrating leukocytes, and preserve oligodendrocyte and neuron number and brain mass. In previous behavioral studies, rats showed improvements in cortical neuronal cell survival and improved cognitive and sensory processing performance after exposure to HI and treatment with IAIPs [36,37,38].

Interventions at or near the time of HI-related brain injury may be effective in reducing injury severity. Even with pharmacological intervention, HI may result in long-term neurodevelopmental abnormalities. However, early life experience and consequent neuroplasticity may improve outcomes. After HI injury in rodent models, several components of the immune system can remain activated for weeks to months, indicating that there may be further adaptive immune responses after the initial insult [30]. Simultaneously, HI-related brain injury can predispose one to long-term cognitive and behavioral deficits, including impaired working memory [32,37,39,40]. Similar cognitive and behavioral abnormalities have been observed in humans with diagnosed or presumed HI or full term infants with HIE, but outcomes vary widely [41,42,43]. Even full term infants with mild HIE, who may not qualify for treatment with hypothermia or follow-up, exhibit cognitive and behavioral impairment including lower IQs, compared with their unimpaired peers [44,45]. Early infant, childhood, and school age cognitive/behavioral interventions significantly contribute to attenuating developmental abnormalities (Reviewed in [46,47]). Studies of early experience and enrichment, including models of enriched housing and increased activity during pre-weaning and juvenile periods, indicate improvement in performance in various models of developmental brain injury [48,49,50,51]. Cognitive and behavioral training along with therapy, particularly early and continuous interventions with strong parental involvement, have similarly been shown to improve long-term cognitive outcomes in pre-term infants and those with high-risk birth complications [52]. Previous research from our group has shown that early life experience with spatial or non-spatial learning resulted in stable long-term improvements in adult learning [36]. Similar long-term improvements have been observed in auditory sensory processing performance after early life experience [53].

The radial-arm water maze (RWM) paradigm used in this study facilitates the ability to assess working memory along with providing an increased cognitive load in rats because the task becomes increasingly difficult as the platforms are removed across trials [39,54,55]. Furthermore, this task provides intrinsic motivation for goal acquisition (escape) and thus, does not require pre-training or food deprivation to facilitate task engagement and learning. The learning dynamics explored in this paradigm have direct translational implications for working memory capacity in humans, which can be negatively impacted by HI related brain injury [56,57]. Previously, we showed that early life (P33+) RWM experience, using a higher demand version of the present paradigm, resulted in significant long-term improvement in learning performance in neonatal rats exposed to HI and treated with IAIPs as compared to HI-exposed vehicle-treated animals assessed only as adults [37]. These findings provided initial evidence that early domain-specific cognitive experiences combined with anti-inflammatory agents improve long-term recovery potential.

The purpose of the current study was to examine the combined effects of early treatment with IAIPs and working memory domain-specific training in juvenile (P31–49) and adolescent (P52–70) rats with HI injury using the RWM. In addition, we examined the long-term benefits of these early experience windows by evaluating adult (P73+) retesting outcomes across the two age groups. The present study expands our previous work to compare differences in age at first experience (P31+ and P52+) and retention (P73+) of working memory performance to identify an optimal window for cognitive intervention and improved outcomes [37].

## 2. Materials and Methods

### 2.1. Animal Subjects

All procedures were approved by the Regis College Institutional Animal Care and Use Committee in compliance with the NIH Office of Animal Laboratory Welfare (Regis IACUC protocol #03.1-2017, approved 20 March 2017, attached to grant 2R15HD077544). Fourteen Timed-pregnant Wistar rat dams (Charles River Laboratories, Cambridge, MA, USA) were delivered to Regis College on embryonic day six, 15 days prior to birth. On postnatal day (P1), pups were separated into litters of eight males and two females to control for litter size and sex ratio. The present studies were performed in male subjects because our research group and others have shown that HI injured males exhibit significant cognitive impairments as compared with females with the same injury. This is an observation that is consistent with reports of HI related brain injury in rodents and human premature infants [58,59,60].

### 2.2. Surgery and Treatment

The induction of HI injury followed a modified Rice-Vanucci protocol, identical to the surgeries and administration of IAIPs performed in our previous experiments [29,36,37,38]. On P7, rats were weighed and randomly assigned to Sham, hypoxia-ischemia (HI), and HI + IAIPs treatment groups. Rats were coded at this time for tracking. Body temperature was maintained before and after surgery using Deltaphase isothermal heating pads (Braintree Scientific, Inc., Braintree, MA, USA) heated to liquid phase, 37 °C. During surgery, animals were stabilized on a surgical surface warmed by a heating pad (37 °C) and anesthetized using 3–4% isoflurane vapor (1–2% maintenance). The right common carotid artery (RCCA) was located and completely cauterized following a 1 cm midline incision on the ventral surface of the neck. After surgery, pups were returned to their dams for one hour. After this period, subjects were given the first injection of IAIPs (30 mg/kg) or equivalent volume of saline (vehicle) depending on treatment group assignment (HI + IAIPs, HI, or Sham). Purified human IAIPs were obtained from author YPL (ProThera Biologics, Inc., Providence, RI, USA; as previously in [36,38]). Because the initiating event in perinatal HIE is clinically definable and subsequent inflammatory injury occurs over the course of days, pre-clinical models have used this window after initial injury for testing of interventions [30,36,37,38]. Immediately after injection, HI subjects were placed on a heating pad (37 °C) in a sealed chamber and exposed to a hypoxic (8%), humidified atmosphere for 120 min. Sham subjects received only an incision, sutures, saline injection, and time away from dams (120 min) in a warm (37 °C), open-air container. Temperature was monitored via infrared laser thermometer (RYOBI Limited) to ensure stability throughout surgical and hypoxia procedures. All subjects were reweighed and given either saline or IAIP injection (30 mg/kg) 24 h after the initial dose and returned to their dams. Pups remained with the dam until weaning and then were pair housed with a litter mate on postnatal day 21. Body weight was monitored at regular intervals until weaning (P21) and again near the end of each RWM trial (P49, P78, P88) across the study. Three HI-exposed rats did not survive the protocols. Therefore, the final treatment numbers (*n*) for the study were as follows, Sham *n* = 22, HI *n* = 23, HI + IAIPs *n* = 25.

### 2.3. Eight Arm Radial Water Maze (RWM)

All behavioral procedures were conducted on coded animals without knowledge of the treatment group. Eight arm water maze (RWM) testing was similar to that previously described [37,39] and included the same testing area and layout of visual cues and researcher positions. At weaning, subjects from each treatment group were randomly assigned to either juvenile (P31+) or adolescent (P52+) experience groups, which determined the age of initial maze experience (Figure 1). Platform and starting arm locations remain fixed and associated with the same extra-maze cues throughout testing across all days. Each subject was given 120 s to find one of the escape platforms after entry into the maze. Each trial was followed by a 60 s delay in which the animal was placed in a warmed (37 °C) holding cage (to maintain body temperature) in preparation for the subsequent trial. Between trials (60 s), the previously located platform was removed. Testing was performed five days a week over a five-week interval with a three-week break between the third and fifth weeks (Figure 1). Arm entries and the platform used for escape were recorded during each trial (See [39] for detailed procedure). Working Memory Incorrect errors (WMI) are defined as repeat entries into an arm that never contained a platform. The primary dependent behavioral variable was WMI errors because previous studies show that this type of error reflects a moderate level of task demand capable of revealing more subtle injury, age-related cognitive impairments and stepwise learning in rats [37,61].

### 2.4. Body and Brain Weight

Body weight was measured at the following study ages: P7, P8, P12, P15, P18, P21, P49, P78, and P88. Brains were collected at the end of the second session of testing (P88 for juvenile experience group and P109 for adolescent experience group). At the end of session two, for each experience group, subjects were deeply anesthetized using 100 mg/kg of pentobarbital and euthanized via cardiac perfusion with 200 mL of 0.9% saline solution. Perfused brains were removed, blocked in the coronal plane (approximately +5.0 anterior to Bregma and −7.5 posterior to Bregma), and weighed. The brains were preserved for future histological analysis. Researchers were not aware of the treatment groups until postmortem analysis when subject treatment codes were revealed.

### 2.5. Statistics

Statistics were performed using SPSS Statistics software (v24, IBM, Armonk, NY, USA). The experimental n is supported by previous studies and a power analysis obtained from previous data, showing that the expected behavioral effect size required a minimum n of 9 per treatment condition to achieve significance at the 0.05 (two-tail) level with a minimum power of 0.8 [37,39]. The numbers for each treatment and experience group were selected as optimal minimums. All errors were totaled for the behavioral analysis, separated by trial and week (4 Trials; 5 Weeks; within-subject factors), averaged by HI treatment and experience group (Sham, HI, HI + IAIPs Treatment; Juvenile and Adolescent Experience; between subject factors), and repeated-measures analysis of variance (RMANOVA) were performed (SPSS). Brain and body weights were compared using the same between subject factors; repeated measures for body weights were based on age at weight determination. Overall statistical comparisons are displayed in text as F [degrees freedom factor, degrees freedom error] and *p* values. Specific behavioral patterns for all treatment groups over the course of this experiment were expected based on previous studies using the same learning paradigm. Error rates were specifically expected to increase as trials progressed with fewer platform targets leading to a higher cognitive demand. We also expected to observe lower WMI error rates across weeks of testing as a function of learning. Only the second, third, and fourth trials were included in the post-hoc analyses based upon the expected trial demand progression when overall RMANOVA statistics showed a trial x treatment interaction. Pairwise comparisons of simple mean differences were performed for statistical groupings of two or three factors with Sidak corrections for multiple repeated measure comparisons of groupings when statistical main effects were observed using RMANOVA. Post-hoc pairwise comparisons are presented as a *t*-test (comparison, t [absolute mean difference], *p* value). Differences are shown on graphs as mean ± SEM, * indicates *p* < 0.05 for significant pairwise differences or homologous groupings per Tukey post-hoc analyses. All SPSS code and data used are provided in Appendix A.

## 3. Results

### 3.1. HI Exposure Resulted in Main Effects of Body and Brain Weight

A Treatment (three levels; Sham, HI, HI + IAIPs) × age (nine levels; P7, P8, P12, P15, P18, P21, P49, P78, P88, within group) × Experience (two levels; Juvenile, adolescent) repeated measure ANOVA revealed a main effect of Treatment on body weight (F(2,63) = 37.87, *p* < 0.001). Both HI and HI + IAIPs treatment groups had lower body weights as compared to Sham subjects across the duration of the study. Experience group was not observed to affect body weight (Figure 2a). After tissue collection, brain injury was visually confirmed in HI subjects and the brains were weighed. A one-way ANOVA for brain weight across the three treatment groups revealed a significant effect of Treatment (F(2,63) = 37.87, *p* < 0.001) with both HI and HI + IAIPs treatment groups demonstrating significantly lower brain weights as compared to Shams, corrected for multiple comparisons (Figure 2b). Although timing of study completion and tissue collection varied across testing groups (G1, P88 and G2, P109, respectively), both groups had reached adulthood at the time of tissue collection. Differences in brain weight were not observed as a function of Experience. Therefore, differences in age of sacrifice across experience groups did not affect brain weight outcome.

### 3.2. HI Exposure Increased Error Rates and Juvenile Testing Improved Performance on Retest

A Treatment (three levels, between group; Sham, HI, HI + IAIPs) × Experience (two levels, between group; Juvenile, Adolescent) × Week (5 weeks, within group) × Trial (four trials, within group) RMANOVA revealed several main effects and interactions for WMI errors (Table 1 and Table 2). Firstly, there was a main effect of Treatment (F(2,61) = 11.81, *p* < 0.001), with Sham subjects making fewer errors on average across all weeks of testing as compared to HI-exposed and IAIP-treated subjects (Figure 3a). This factorial analysis also showed an effect of Week (F(4,244) = 15.15, *p* < 0.001), with all subjects making fewer errors as the weeks progressed. Similarly, an effect of Trial (F(3,183) = 386.09, *p* < 0.001) was also observed, indicating that as the trials progressed, error rates increased. Experience group did not have an overall effect on performance (F(1,61) = 1.522, *p* = 0.222), indicating that juvenile and adolescent experience groups did not show an overall difference in error rates across treatment groups.

In addition to these main effects, results showed a Trial × Treatment interaction (F(6,183) = 12.58, *p* < 0.001) (Table 2), demonstrating that as the trials progressed, error rates for the two treatment groups changed, with the sham subjects making fewer errors than either HI exposed group across all five weeks (Figure 3b). Analyzing simple means with Sidak correction, WMI errors increased as expected for later trials and the overall Treatment effect was significant in trials three (Sham vs. HI, *t*[2.994], *p* = 0.001; Sham vs. HI + IAIPs, *t*[3.001], *p* < 0.001) and four (Sham vs. HI, *t*[6.852], *p* < 0.001; Sham vs. HI + IAIPs, *t*[6.420], *p* < 0.001). This difference in later trials emphasizes the importance of the increased cognitive load associated with the successive trials.

Because age has the potential to affect performance at first experience, juvenile (P31) and adolescent (P52) groups were compared for overall differences in performance and for differences across specific trials and weeks. Several interactions were observed for juvenile versus adolescent experience groups (Table 2). Although there was no overall group effect, there was a Trial × Experience interaction (F(3,183) = 2.96, *p* = 0.034), indicating that at least one trial exhibited a differential response based on experience group alone. Even though the largest difference between the experience groups appears on trial 4 (Figure 4a), this does not reach statistical significance (juvenile vs. adolescent, *t*[1.973], *p* = 0.093), indicating that there was additional interaction influencing the overall comparison. Specifically, a Week × Experience interaction is also seen (F(4,244) = 2.73, *p* = 0.030)) and this was best observed with a significant difference in WMI errors between the juvenile and adolescent experience groups in week 4, with juvenile experience subjects performing better (making fewer errors) as compared to their adolescent counterparts (juvenile vs. adolescent, *t*[2.533], *p* = 0.017; Figure 4b).

### 3.3. IAIP Treatment Improved Performance in Juvenile Testing. Juvenile Testing Improved Performance on Trials with High Cognitive Demand and Reduced Working Memory Errors during Retest

Further comparisons of WMI errors were made for treatment and experience groups at individual trials and weeks because of the above overall effects and interactions, analyzing separately simple mean comparisons of Treatment × Experience × Week × Trial (Figure 5). We controlled for multiple comparisons using Sidak corrections. Subjects treated with IAIPs showed similar performance to the Sham subjects during the first week of juvenile experience testing. However, the HI-exposed animals performed significantly worse than Sham in the third and fourth trial of the first week (Sham vs. HI, *t*[5.418], *p* = 0.024; *t*[10.373], *p* = 0.016, respectively), the second trial of the second week (Sham vs. HI, 3.655, *p* = 0.011), and the third trial of the second week (Sham vs. HI, *t*[5.555], *p* = 0.002). In comparison, IAIP-treated subjects performed worse than Sham on week 2 trial 4 (Sham vs. HI + IAIPs, *t*[8.734], *p* = 0.007) and week 5 trial 3 (Sham vs. HI + IAIPs, *t*[4.874], *p* = 0.012) when cognitive demand was high. For these trials, HI exposure did not differ significantly from the Sham subjects. In contrast, HI exposed subjects in the adolescent experience group made significantly more errors as compared to Shams on week 2 trial 2 (Sham vs. HI, *t*[3.073], *p* = 0.04) and week 4 trial 4 (Sham vs. HI, *t*[10.673], *p* = 0.045), with HI+IAIP-treated subjects performing similar to HI subjects. IAIP-treated subjects performed worse than Sham on week 2 trial 4 (Sham vs. HI + IAIPs, *t*[7.636], *p* = 0.025) and HI-exposed IAIP-treated subjects performed worse than Sham on week 3 trial 4 (Sham vs. HI, *t*[9.045], *p* = 0.018; Sham vs. HI + IAIPs, *t*[9.462], *p* = 0.008).

Examination of the experience groups revealed specific improvements made by the juvenile testing group as compared to the adolescent group. Specifically, Experience × Week × Trial interactions revealed that the two experience groups showed a dramatic difference in error rates on trial 4 of week 4 (juvenile vs. adolescent, *t*[6.663], *p* = 0.007). Further, HI exposed animals with juvenile RWM experience made significantly fewer errors on week 4 trial 4 than those with the adolescent experience (HI juvenile vs. HI adolescent, *t*[11.200], *p* = 0.013; Figure 5d,i).

## 4. Discussion

The goals of the current study were to expand upon the pharmacological and behavioral interventional strategies previously shown to improve performance in rats exposed to HI injury and investigate age windows for cognitive experience to improve long-term outcomes. The eight arm Radial Water Maze (RWM) paradigm closely resembles that used in previous studies of working memory in rats and humans [57,62]. Both the Sham and injured subjects showed expected and significant improvement over multiple weeks of domain-specific working memory testing [54]. However, significant differences in performance were observed as a result of both pharmacological treatment with IAIPs and age of testing experience. Important significant interactions were also observed across weeks of testing, testing trials, and upon retesting.

Overall, hypoxia-ischemia resulted in significant impairment in working memory performance for both juvenile and adolescent experience groups across all five weeks of testing and all trials as compared to shams. Specifically, HI+IAIP treatment resulted in an average of 5.9 errors per trial per week as compared to 6.2 errors for HI exposed subjects, a 4.8% difference when data were collapsed for all weeks, trials, and age at first test. A similar result was observed for specific trials (trials 1–4) across all weeks and both experience groups (juvenile and adolescent). In these examples, HI and HI+IAIPs treatment groups did not differ. Nonetheless, several significant interactions, including Trial by Treatment, Trial by Experience, and Week by Experience, revealed improved IAIP-treated performance in conjunction with the age of initial experiential learning (juvenile vs. adolescent) within specific weeks of examination. In particular, HI+IAIP-treated subjects made an average of 6.7 errors as compared to 9.6 errors for HI-exposed subjects, a 30% advantage of IAIPs treatment over HI performance in the first week of juvenile testing, by the third trial. This divergence was even more apparent on the fourth trial of week one, with HI+IAIP-treated subjects averaging 15.2 errors as compared to HI exposed subjects averaging 22.1. This contrast reflects a 31% advantage of IAIP-treated over HI subjects. Although not significantly different by direct comparison, the performance of HI+IAIP-treated subjects was closer to that of Shams (11.7 errors) than to subjects exposed to HI at this testing interval. Importantly, performance improvements in juvenile subjects treated with IAIPs were maintained during much of the second week of testing (trials 2 and 3). Improvement in performance observed in juvenile subjects treated with IAIPs as compared to HI animals was not observed in the adolescent experience group. Differences between HI exposed and HI+IAIP-treated subjects were much smaller in the adolescent experience group as compared to the juvenile window. For example, on week 2 trial 4, HI+IAIP-treated and HI-exposed adolescent subjects made 12.3 versus 13 WMI errors on average, respectively. Although directionality of the comparison remained, the difference in error rates between the groups was 8.1%. These findings are largely consistent with previous work [37]. However, the shorter inter-trial interval in the current study (60 s) as compared to Gaudet et al. (90 s) [37] may have decreased cognitive demand enough to shift therapeutic effects of IAIPs to early weeks of testing. Nevertheless, findings from both studies suggest that IAIP treatment may improve working memory performance most effectively after HI injury when combined with repeated early life (juvenile P31+) experience.

Comparison of earlier (juvenile, adolescent) and later (adult) maze performance revealed a period of increased error rates during week 4, after the 3-week break. In week four, all subjects regressed to error rates observed in the first two weeks of testing. However, despite this deterioration in performance, subjects with earlier juvenile experience made significantly fewer errors in week 4 testing, after the break, as compared to the adolescent group, regardless of treatment or HI-exposure. The juvenile experience group made an average of 5.1 errors as compared with the adolescent experience group, which made 7.7 errors. Therefore, the juvenile experience group retained their improved performance from week 3, prior to the break. In contrast, the adolescent group provided the main contribution to a perceived overall regression in performance after the break (week 4). In particular, juveniles made an average of 3.9 errors during week three. In contrast, the adolescent group performed similar to week 1 during the week 4 retest, averaging 6.6 errors. These findings suggest a complete lack of retention in week 4 for the adolescent group. These results support the hypothesis that early (juvenile P31–49) domain-specific cognitive experience can improve long-term performance in normal animals and those with developmental brain injury. The greater cognitive demand of the later trials reflects larger differences in error rates between juvenile and adolescent groups on trial 4 of week 4 (averaging 12.8 errors versus 19.4, respectively). Both Sham and HI+IAIP-treated juvenile subjects showed improved performance on trial 4 of week 4. However, the largest overall improvement was observed in the juvenile HI-exposed subjects, which made an average of 12.2 errors as compared to 23.4 in the adolescent experience group. The contrast between HI subjects at the two ages and experience groups reflects a 47.9% difference. This striking difference indicates significant benefits of early repeated working memory experiences, even for HI animals without pharmacological intervention.

These findings expand our understanding of combining pharmacological and behavioral interventions and identify an apparent window of opportunity in which early working memory domain-specific experiences can improve long-term learning performance after HI related brain injury. A small number of studies also suggest a sensitive period for learning around P30 in rats [50,63]. These studies have shown significant increases in dendritic arborization and spine density associated with simple exposure to working memory tasks and other enrichment modalities. Not surprisingly, a decline in cognitive performance has been demonstrated between young and old animals [55,64]. However, in many such studies, the difference between “young” (often ~3 months) and “old” (often ~18 months) is vastly different as compared to the juvenile (P31) and adolescent (P52) age windows investigated here. Few studies have investigated cognitive function and learning longitudinally in this manner, i.e., explicitly testing a period of inactivity after a cognitive or other enriching experience [65]. Our research, including this current study, supports the concept that when task demand is high, longitudinal task presentation [37,38], as opposed to single discrete task presentation, provides a robust measure of long-term cognitive outcome in typical rats and those with neurodevelopmental impairment. However, divergent cognitive and/or sensory processing domains may be sensitive to differences in performance at longer time points than presented here [53]. Nevertheless, it is clear that behaviorally relevant translational models require consideration of task timing and presentation order. These considerations would provide greater support for the extension of findings to the human population, where effects of HI related brain injury are often impacted by varied life experiences and developmental time courses [8,9,41,43].

Prematurity and low birth weight are risk factors for additional perinatal HI-related brain injury and, consequently, can result in neurological impairment and developmental delay [42,66,67,68]. Low weight infants who gain weight earlier during their hospitalizations have more optimal neurodevelopmental outcomes [69]. The weight gain trajectory was significantly lower throughout the current study in both HI groups regardless of treatment and experience. In addition, brain weights were also significantly lower after exposure to HI-related brain injury as compared with the Sham group regardless of treatment or experience condition. Differences in growth rates and body weight gain have been observed as outcomes in HI-related brain injury in rats [70,71] and malnutrition has been shown to have detrimental effects on neurodevelopment after HI-related brain injury [72]. Some studies have monitored body weight gain and provided nutritional supplements to reduce mortality after HI-related brain injury [73]. Like our current study, recent studies have noted differences in weight gain and brain weight can occur rapidly after HI-related brain injury [35]. In contrast, Borjini and colleagues [74] reported no difference in body weight between Sham and HI-exposed rats, potentially reflecting different study protocols. Interestingly, animal strain (Wistar rats), housing conditions, and surgical and feeding protocols in the present study were comparable to those reported by Borjini et al. [74]. However, Borjini and colleagues [74] utilized 60 min of hypoxia compared with 120 min in our study, which likely accounts for our observed body weight differences. Furthermore, it is not clear whether litter size was controlled in the study by the Borjini group. Comparisons of our average neonatal rat weight on P7 (16.7 g) with the range of weights reported by Borjini (12–14 g) suggests that our animals were initially heavier. These initial weight differences are potentially critical in the projections of weight gain over the study periods. Chronic differences in overall weight are correlated with stress, diet, and physical activity and have varying effects on cognitive and physical performance in memory tasks [75,76,77]. As weight is related to overall health, development, and enrichment, future longitudinal studies should endeavor to publish weight and other growth relevant measures.

Currently, preclinical studies are focused on the application of immunomodulation or anti-inflammatory molecules as candidates to attenuate hypoxic-ischemic and other forms of brain injury. Neonatal rats treated with IAIPs immediately before hypoxia and 24 h later show comparable error rates to Sham subjects during the first two weeks of juvenile testing. In contrast, HI-exposed subjects showed significantly more errors as compared to Sham subjects during this period. IAIPs have previously been shown to be neuroprotective in preclinical studies of HI-related brain injury, improving sensory and behavioral function [33,37,38]. Of note, multiple phases of HI-related injury progression and recovery from injury have been documented (reviewed in Hagberg [78]). Some of the neuroplastic responses to injury, including glial scarring, clearance of necrotic, and underdeveloped tissue, and development of alternative neural tracks, cortical atrophy, and loss of brain mass progress for weeks to months. Further, these processes are highly dependent on inflammatory responses of the brain [30,79,80]. It was during the early stages of inflammation that IAIPs were given in the current study. The primary effects seen following IAIP treatment in our behavioral tests were several weeks removed from the initial HI-related brain injury. Therefore, treatment may have altered the trajectory of these inflammatory responses. Specific neuroprotective mechanisms of IAIPs are thought to involve systematic repair and maintenance of cerebral extracellular matrix, the blood-brain barrier, and cerebrospinal fluid system [81]. Despite numerous studies showing functional improvement in a range of cognitive and sensory processes affected by HI related injury, further mechanisms by which IAIPs may be neuroprotective have yet to be identified. However, pre-clinical responses are promising for IAIPs and our findings suggest that the combination of pharmacological treatment with early cognitive experience will provide optimal outcomes.

## 5. Conclusions

Our findings reveal improvements in juvenile (P31+) working memory performance for rats treated with IAIPs as compared to HI exposed counterparts. In addition, juvenile RWM experience resulted in significant reductions in error rates as compared to adolescent (P52+) experience, indicating long-term cognitive benefits of early experience regardless of injury or treatment type. The findings provide additional support for the role of both domain-specific cognitive and pharmacological interventions to mediate the short- and long-term impact of neonatal brain injury.

## Figures and Tables

**Figure 1 brainsci-10-00999-f001:**
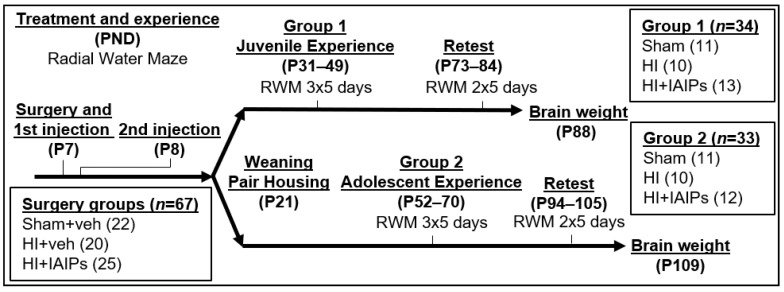
Timeline for surgery, exposure to Hypoxic-Ischemic (HI), Inter-alpha Inhibitor Proteins (IAIP) treatment, Radial Water Maze (RWM) experience, and tissue collection by post-natal day (P#). Subject numbers post-surgery separated into experience groups (Group 1 = Juvenile, Group 2 = Adolescent) are indicated in boxes. All subjects were male rats.

**Figure 2 brainsci-10-00999-f002:**
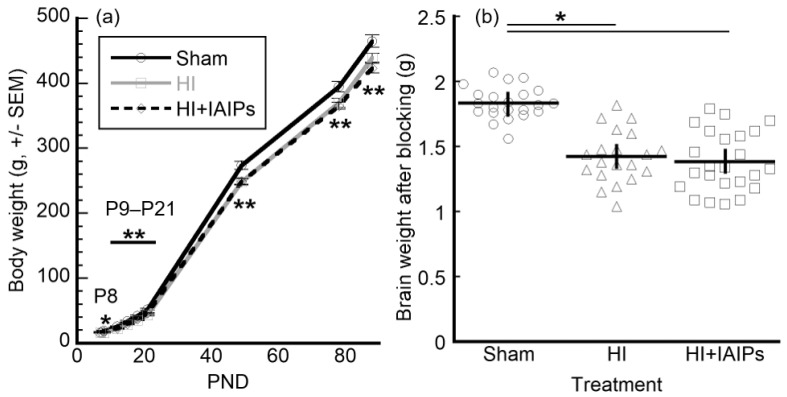
(**a**) Body weight of animals (±SEM) over time of study. Total *n* = 67 after surgery, Sham (22), HI (20), HI + IAIPs (25). Repeated measures ANOVA indicates an overall effect of HI injury on body weight, * indicates that HI is significantly lighter than Sham and ** indicates both HI and HI + IAIP cohorts are significantly lighter than Sham animals. (**b**) Brain weights for Sham, HI, and HI + IAIPs subjects at time of collection. Horizontal bars are mean values and vertical bars indicate SEM. Total *n* = 66 after brain collection, Sham (22), HI (20), HI + IAIPs (24). ANOVA with Tukey’s homologous subsets indicate that brain weights were significantly lower in both injured treatment groups as compared to shams. HI = Hypoxic-Ischemic injury; IAIPs = Inter-alpha Inhibitory Proteins; PND = post-natal day.

**Figure 3 brainsci-10-00999-f003:**
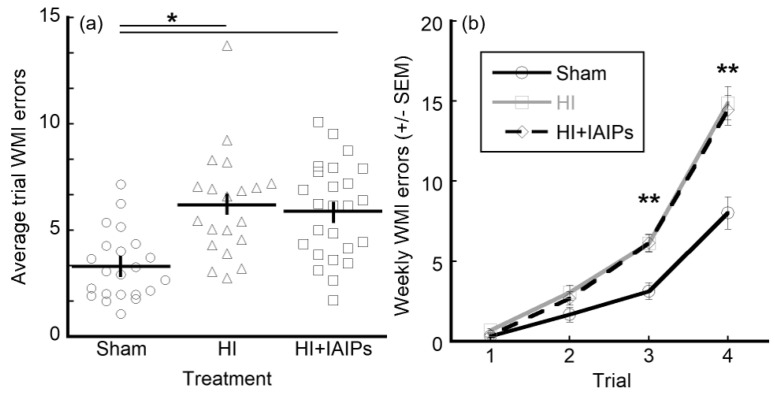
Treatment effects on WMI errors (±SEM), a moderate cognitive demand task component. Total *n* = 67, Sham (22), HI (20), HI + IAIPs (25). Horizontal bars indicate mean and vertical error bars represent SEM. ANOVA and post-hoc tests indicate both HI and HI + IAIPs treatment groups performed worse than Shams in overall WMI errors across all trials and weeks (**a**) * indicate significant differences by Tukey’s homologous subsets and (**b**) ** indicates both HI and HI + IAIPs performed significantly worse than Sham in later trials of the maze protocol over all weeks by paired comparison with Sidak correction.

**Figure 4 brainsci-10-00999-f004:**
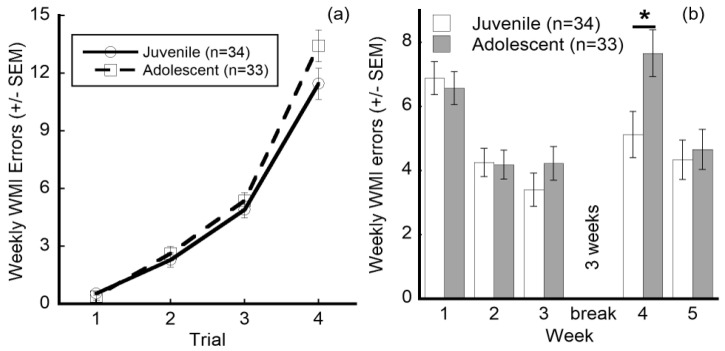
Effects of juvenile versus adolescent experience on performance. (**a**) Interaction of experience group and trial WMI error performance (±SEM). ANOVA indicates a significant difference between experience groups at specific trials. (**b**) Interaction of experience group and week WMI performance (±SEM). * Indicates a significant paired comparison in performance overall between the experience groups at this week, with Sidak correction for multiple comparisons.

**Figure 5 brainsci-10-00999-f005:**
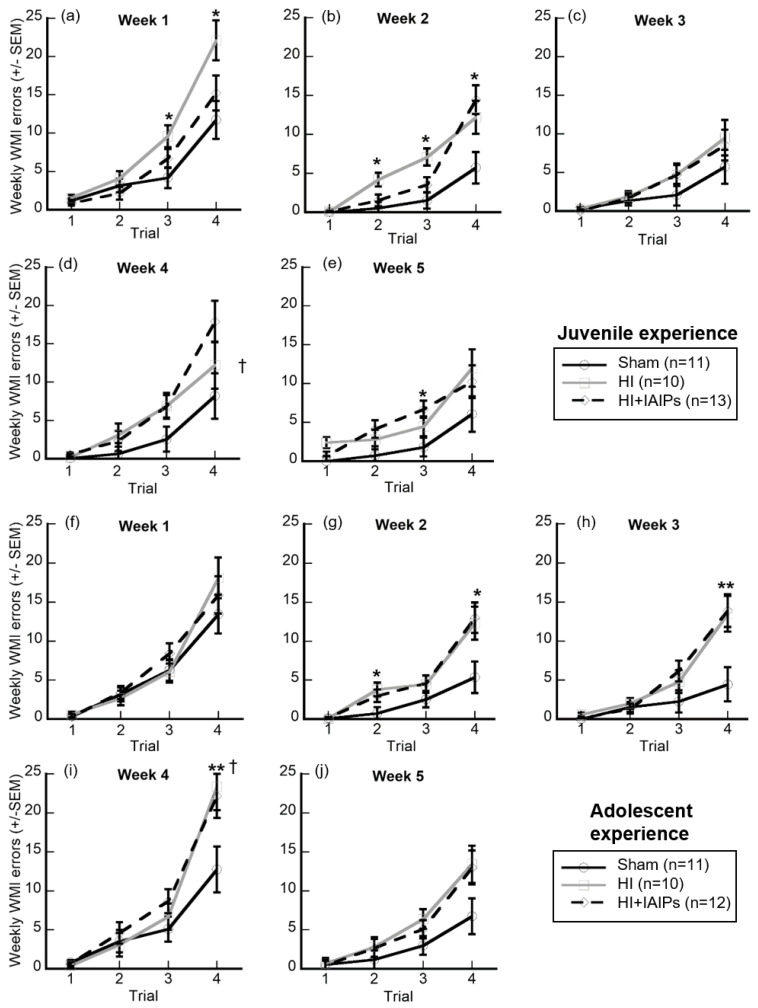
Expanded representation of WMI error performance (±SEM) on individual trials collapsed across weeks. Top two rows (**a**–**e**): juvenile first RWM experience starting at P31. Total *n* = 34, Sham (11), HI (10), HI + IAIPs (13). Bottom two rows (**f**–**j**): adolescent first RWM experience starting at P52. Total *n* = 33, Sham (11), HI (10), HI + IAIPs (12). Each plot is aligned by that experience group’s number of weeks of experience. A three-week break occurs between Week 3 and Week 4 for both experience groups. Paired comparisons were made between both treatments within each graph and between experience groups at the same relative trial, for example, Figure 5a,f. * indicates a trial where one injured treatment group, either HI or HI + IAIPs, performed significantly worse than Sham. ** indicates a trial where both HI and HI+IAIP treatment groups performed worse than Sham. † indicates a trial where the HI-treated, juvenile experience cohort (**d**) significantly differs from the HI-treated, adolescent experience cohort (**i**). All corrected by Sidak correction for multiple comparisons.

**Table 1 brainsci-10-00999-t001:** Main and interaction effect (repeated measures ANOVA) and simple means comparisons (*t*-tests) of RWM errors between group factors. Sidak corrections used for *t*-tests with multiple comparisons. (* for a value indicates a significant paired difference, see significance column, * in a column title indicates a significant overall or interaction effect, as per RMANOVA.).

Factor (*n*)	Average WMI Errors		Significance [Test, *p*]
Exposure/Treatment			* Main effect [RMANOVA, *p* < 0.001]
Sham + Vehicle (*n* = 22)	3.289		
HI + Vehicle (*n* = 20)	6.188 *		* vs. Sham [*t*-test, *p* < 0.001]
HI + IAIPs (*n* = 25)	5.909 *		* vs. Sham [*t*-test, *p* < 0.001]
Experience Group			NS Main effect [RMANOVA, *p* > 0.05]
Juvenile (*n* = 34)	4.798		
Adolescent (*n* = 33)	5.458		
Exposure × Treatment Interaction	Juvenile	Adolescent	NS Interaction effect [RMANOVA, *p* > 0.05]
Sham + Vehicle	2.882 (*n* = 11)	3.695 (*n* = 11)	
HI + Vehicle	6.075 * (*n* = 10)	6.300 * (*n* = 10)	* vs. Sham [*t*-test, *p* < 0.001]
HI + IAIPs	5.438 * (*n* = 13)	6.379 * (*n* = 12)	* vs. Sham [*t*-test, *p* < 0.001]

**Table 2 brainsci-10-00999-t002:** Multifactorial interaction comparisons (repeated measure ANOVA) and simple means comparisons (*t*-tests) of RWM errors. Sidak corrections used for *t*-tests with multiple comparisons. (* for a value indicates a significant paired difference, see significance column, * in a column title indicates a significant overall or interaction effect, as per RMANOVA.).

Factor 1 (*n*)	Factor 2 Average WMI Errors					Significance [Test, *p*]
Exposure/Treatment	Trial					* Interaction effect [RMANOVA, *p* < 0.001]
	1	2	3	4		
Sham + Vehicle (*n* = 22)	0.336	1.664	3.136	8.018		
HI + Vehicle (*n* = 20)	0.700	3.050	6.130 *	14.870 *		* vs. Sham in trials 3 and 4 [*t*-tests, *p* < 0.001]
HI + IAIPs (*n* = 25)	0.374	2.685	6.138 *	14.438 *		* vs. Sham in trials 3 and 4 [*t*-tests, *p* < 0.001]
Experience Group	Trial					* Interaction effect [RMANOVA, *p* < 0.05]
	1	2	3	4		
Juvenile (*n* = 34)	0.548	2.293	4.896	11.456		
Adolescent (*n* = 33)	0.392	2.639	5.373	13.428		NS [*t*-tests, *p* > 0.05]
Experience Group (total)	Week					* Interaction effect [RMANOVA, *p* < 0.001]
	1	2	3	4	5	
Juvenile (*n* = 34)	6.883	4.253	3.398	5.122	4.336	
Adolescent (*n* = 33)	6.571	4.181	4.223	7.655 *	4.660	* vs. juvenile in wk4 [*t*-tests, *p* < 0.05]
Sham + Vehicle						
Juvenile (*n* = 11)	5.068	1.955	2.364	2.863	2.159	
Adolescent (*n* = 11)	5.818	2.182	2.068	5.545 *	2.864	* vs. juvenile in wk4 [*t*-tests, *p* < 0.05]
HI + Vehicle						
Juvenile (*n* = 10)	9.350	5.900	4.100	5.600	5.425	
Adolescent (*n* = 10)	6.875	5.175	5.225	8.400 *	5.825	* vs. juvenile in wk4 [*t*-tests, *p* < 0.05]
HI + IAIPs						
Juvenile (*n* = 13)	6.231	4.903	3.731	6.903	5.423	
Adolescent (*n* = 12)	7.021	5.188	5.375	9.021 *	5.291	* vs. juvenile in wk4 [*t*-tests, *p* < 0.05]

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
