# Peer review of "Effects of Juvenile or Adolescent Working Memory Experience and Inter-Alpha Inhibitor Protein Treatment after Neonatal Hypoxia-Ischemia"

_brainsci, 2020, doi:10.3390/brainsci10120999_

Round 1

Reviewer 1 Report

This is an interesting study with a focus on hypoxic /ischemic brain injury in the neonate.

The study of IAIPs and multiple weeks of RWM assessment provide the opportunity for obtaining results with future clinically relevant implications.

The data is clearly presented and the interpretation is appropriate.

While HI injury impaired performance the IAIP treated HI animals tested as juveniles made fewer errors than untreated HI animals. Combining behavioral and anti-inflammatory intervention in HI improved subsequent learning performance.

Key findings were that IAIP treated groups made fewer errors for HI-exposed subjects in the first week of juvenile testing, by the third trial. This difference was even greater on the fourth trial of week one. Performance improvements in juveniles with IAIP treatment were maintained during much of week two of testing (trials 2 and 3). In adolescence differences were much smaller. Hence IAIP treatment may improve working memory most effectively after HI injury under conditions of juvenile P31+ experience.

The study points to a possible window in which early working memory experiences can improve long term learning performance after HI related brain injury.

Reviewer 2 Report

This is an excellent study by a well-established group, and addresses an important clinical issue. I believe it will be an excellent addition to Brain Sciences. My comments/concerns for improvement are minor.

1. Although described in Methods, it may be useful to the casual reader to include M (for males) in the study design table (e.g., Group 1 (n=34 M)).

2. It may be useful to refer to the HI+IAIP group as such, rather than just IAIP, in Figure 1 groups and in the Results section. This is because no IAIP-sham group is included. This would clarify understanding of the Results section.

3. Consistently capitalize (or not) statistical variables (e.g., treatment) in Results. It is capitalized once in the first sentence of Results (line 208). I personally prefer to see variables capitalized but either way, a consistent rule should be applied.

4. The possibility of Experience differences due to different ages at Retest should be considered (line 255-262). This seems unlikely given that the superior performance was seen in the animals retested at an earlier age, but should be mentioned as a possible confound nonetheless.

5. The authors note that timing differences in post-injury inflammatory expression could interact with age at initial learning experience. Some additional information on key post-injury events at 24 days post injury (group 1) versus 45 days post-injury (group 2) would be informative (lines 410-415).

6. Additional detail regarding the consistency between the current pattern of results and those from the author’s prior work (reference 35. Gaudet, C.M., Lim, Y.-P., Stonestreet, B.S., Threlkeld, S.W. Effects of age, experience and inter-alpha 537 inhibitor proteins on working memory and neuronal plasticity after neonatal hypoxia-ischemia. Behav Brain 538 Res 2016, 302, 88-99. doi: 10.1148/rg.282075066), should be provided in the Discussion, including comparability of the effect size for gains observed with IAIP treatment.

Reviewer 3 Report

The authors investigate the effect of juvenile (P31+) versus adolescent (P52+) memory experience on a rat model of HI at P7 with and with inter-alpha inhibitor proteins injection.

P7 in rat is equivalent to late preterm where hypothermia is not the standard of treatment. IAIP as anti-inflammatory provides an alternate therapy for this group and adjunct therapy for neonatal HIE.

This study is novel as it addresses training over weeks contributes to neurodevelopmental outcome and not just spot training and testing.

1-Introduction:

a-Long but informative. Please elaborate on what is known about IAIP mechanism of neuroprotection. The authors mentioned that it improves cortical neuron cell survival. How does it do that?

2-Methods:

a-Temperature affects outcomes of HI. Was temperature of rats measured or were they just placed on Deltaphase isothermal heating pads?

b- Was there any differences noted between sham and sham + IAIP?

3-Results:

  • Please transform bar graphs to dot plot with mean and SEM in Fig 2 and 3, as they are more informative.
  • In table 1, lumping the junvenile (sham, HI +vehicle and HI + IAIP ) in one group and adolescent (sham, HI +vehicle and HI + IAIP ) in another group would certainly have no significance since the effects of HI in each group is cancelled by the sham. So what is the benefit from this?
  • In table 2, it would be meaningful and more informative to add week 1-5 for each of the (sham, HI +vehicle and HI + IAIP ) with a subset of Juvenile and adolescent.

Exposure/treatment                                       week

                                                                                1              2              3              4              5

Sham+ vehicle (Juvenile) n=11

Sham+ vehicle (adolescent) n=11

HI +vehicle (Juvenile) n=

HI +vehicle (adolescent) n=

HI + IAIP (Juvenile) n=

HI + IAIP (adolescent) n=

d-Fig 5 in informative of how the different groups (sham, HI +vehicle and HI + IAIP )  performed over the weeks. It would make it more clear if there would be comparison between Juvenile and adolescent comparing for example HI week 1 juvenile vs HI week 1 adolescent and HI + IAIP week 1 juvenile vs HI week 1 adolescent and so forth…

4-Discussion:

a-One possible discrepancy between weight and brain weight of HI vs sham is amount of hypoxia. Another possible cause is partial injury to the vagus nerve ( while isolating the carotid artery) which runs along the carotid artery and could affect swallowing and therefore affect weight.

b-This initial IAIP treatment is right after the ischemia and before the hypoxia treatment. Why did the authors choose this instead of after the completion of HI?

c-Please elaborate more on the mechanism of improved neurodevelopmental outcomes associated with IAIP treatment in HI.

Round 2

Reviewer 3 Report

Authors addressed all concerns. I am satisfied with the authors' revision.